# Molecular Mimicry Analyses Unveiled the Human Herpes Simplex and Poxvirus Epitopes as Possible Candidates to Incite Autoimmunity

**DOI:** 10.3390/pathogens11111362

**Published:** 2022-11-16

**Authors:** Sara Begum, Sara Aiman, Shujaat Ahmad, Abdus Samad, Mazen Almehmadi, Mamdouh Allahyani, Abdulelah Aljuaid, Sahib Gul Afridi, Asifullah Khan

**Affiliations:** 1Department of Biochemistry, Abdul Wali Khan University Mardan (AWKUM), Mardan 23200, KPK, Pakistan; 2Faculty of Environmental and Life Sciences, Beijing University of Technology, Beijing 100124, China; 3Department of Pharmacy, Shaheed Benazir Bhutto University Sheringal Dir (Upper), Sheringal Dir 18000, KPK, Pakistan; 4Department of Clinical Laboratory Sciences, College of Applied Medical Sciences, Taif University, P.O. Box 11099, Taif 21944, Saudi Arabia

**Keywords:** molecular mimicry, autoimmune disorders, cross-reactive epitopes, viral infection, sequence and structural homology

## Abstract

Clinical epidemiological studies have reported that viral infections cause autoimmune pathology in humans. Host-pathogen protein sequences and structure-based molecular mimicry cause autoreactive T cells to cross-activate. The aim of the current study was to implement immunoinformatics approaches to infer sequence- and structure-based molecular mimicry between viral and human proteomic datasets. The protein sequences of all the so far known human-infecting viruses were obtained from the VIPR database, and complete human proteome data were retrieved from the NCBI repository. Based on a predefined, stringent threshold of comparative sequence analyses, 24 viral proteins were identified with significant sequence similarity to human proteins. PathDIP identified the enrichment of these homologous proteins in nine metabolic pathways with a *p*-value < 0.0001. Several viral and human mimic epitopes from these homologous proteins were predicted as strong binders of human HLA alleles, with IC_50_ < 50 nM. Downstream molecular docking analyses identified that lead virus-human homologous epitopes feasibly interact with HLA and TLR4 types of immune receptors. The vast majority of these top-hit homolog epitopic peptides belong to the herpes simplex and poxvirus families. These lead epitope biological sequences and 3D structural-based molecular mimicry may be promising for interpreting herpes simplex virus and poxvirus infection-mediated autoimmune disorders in humans.

## 1. Introduction

The human immune system defends against various diseases by recognizing and eliminating pathogens and damaged cells. If this approach is unsuccessful, an elicited immune response is initiated against the body’s healthy cells and tissues, resulting in various autoimmune illnesses (AIDs) [1,2]. Over 100 million individuals worldwide are infected with over 80 distinct types of deadly AIDs [1,3,4,5]. AIDs are caused by a combination of factors, including age, genetics, environment, and microbial infections [6]. Understanding the underlying molecular mechanisms responsible for autoreactive T cells and the pathophysiology of AIDs is critical. The human immune system maintains a delicate equilibrium to distinguish between self- and non-self-antigens. Most autoreactive T cells are normally destroyed. However, only a small fraction survive, and are assumed to be the cause of autoimmune diseases. Although the human immune system has well-established mechanisms for removing or tolerating the autoreactive T and B cells, nonetheless, these cells are commonly activated and cause devastating AIDs [7,8,9]. The T-cell-specific autoimmune illnesses identified in the literature include multiple sclerosis (MS), human type-1 diabetes, type-2 autoimmune hepatitis, meningitis, primary biliary cirrhosis, and autoimmune arthritis [2,10,11]. However, the molecular mechanisms underlying these diseases remain unclear.

One of the key mechanisms that causes autoimmune disorders is molecular mimicry, in which the pathogenic antigens exhibit sequence and structural resemblance to host self-antigens. Viruses are hypothesized to contain antigens that are structurally similar to self-antigens and activate B and T cells, resulting in a cross-reactive response against both self-and non-self antigens, which leads to autoimmunity [12,13,14]. Several viral species have been associated with AIDs. For instance, the herpes simplex virus (HSV) has been reported to be associated with stromal keratitis, while the coxsackie virus has been reported to cause autoimmune myocarditis. Similarly, Theiler’s murine encephalomyelitis virus causes demyelinating illness in animals. Furthermore, diabetic complications are linked to viral infections [15,16,17,18].

In addition to molecular mimicry, there is a mechanism of “bystander activation,” where a non-specific and highly reactive antiviral immune response in a localized pro-inflammatory milieu causes the release of self-antigens from injured tissue. These self-antigens activate autoreactive T lymphocytes, which induce inflammatory responses and contribute to autoimmunity [19,20]. A viral infection triggers the development of new self-antigens and autoreactive T or B cells, which then disseminate to target additional self-epitopes [21,22]. The spread of such epitopes has been described in individuals with rheumatic fever (RF), an autoimmune disease [23].

Bystander activation as well as molecular mimicry pathway involvement have been reported in autoimmune encephalomyelitis (EAE) models of MS, TMEV-IDD, and myasthenia gravis (MG) mediated by West Nile virus (WNV) [24,25,26]. Furthermore, T-cells that react to immune-dominant myelin basic protein (MBP) epitopes may cross-react with certain viral antigens, boosting the possibility of MS [27,28]. Serotype 4 of the Coxsackie B virus (CVB4) infection has been linked to T-cell specific autoimmune type 1 diabetes [29]. In CVB4-induced type 1 diabetes, both bystander activation and molecular mimicry mechanisms have been demonstrated [30,31]. Several studies in human and animal models have examined the role of rotaviruses in the development of autoimmune diabetes, suggesting that autoimmunity may be caused by bystanders [32,33]. Clinical studies have linked influenza infection to diabetes and other pancreatic diseases [34,35,36,37,38,39]. Herpesvirus and Epstein-Barr virus (EBV) infections are reported to be associated with the development of systemic autoimmune diseases (SADs). MS, RA, and Sjögren’s syndrome (SS) are caused by EBV- and HSV-induced autoimmunity [2,40]. Other viruses, including measles, mumps, and rubella, are associated with the development of type 1 diabetes. Some of these viruses can infect and multiply in beta cells, causing autoimmune disorders [41,42].

Numerous hypotheses have been proposed to better understand the underlying processes of virus-induced autoimmunity; however, in most cases, a well-defined specific mechanism remains unclear. Despite the absence of a clear molecular explanation, multiple epidemiological and animal model experiments have shown that a broad spectrum of viruses may cause an autoimmune response. The emergence of autoimmune diseases after viral infection is a complicated process that is affected by several variables, including immune response, infectious dosage, and infection duration [43].

The viral and human proteins and their corresponding epitopes that share sequence and structural homology as well as exhibiting biochemical interactions with major histocompatibility complex (MHC) molecules may activate autoreactive T-cells, which eventually provoke autoimmune diseases. Currently, bioinformatics and immunoinformatics tools have gained considerable attention. These platforms facilitate the understanding of the complexity of peptide binding to different MHC molecules, forecasting cytokine release, and identifying the lead T- and B-cell epitopes [44,45]. We utilized the entire proteome sequences of known human pathogenic virus species from different families, and the data were thoroughly examined in the context of the human proteome to identify potential sequence and structural homologs to prioritize virus-associated autoimmunity candidates. Several viral peptides have shown significant sequence and structural similarities to human peptides, and some of these homologs have been identified as promising T-cell epitopes that may be cross-reactive during the immune response. Knowledge of viral-human homologous proteins and peptides may be useful for understanding the molecular basis of virus-induced autoimmune diseases.

## 2. Results

### 2.1. Non-Paralogous Viral and Human Proteome Sets

The complete proteome sequences of the human pathogenic viruses available in the ViPR database were retrieved. These include viruses belonging to *Paramyxoviridae*, *Caliciviridae*, *Phenuiviridae*, *Flaviviridae*, *Hantaviridae*, *Pogaviridae*, *Hepeviridae*, *Filovirdae*, *Herpesviridae*, *nairoviridae*, *Arenaviridae*, *Peribunyaviridae*, *Poxviridae*, *Rhabdoviridae*, *Reoviridae*, *Coronaviridae*, Dengue viruses, Ebola viruses, Enteroviruses, Lassa viruses, and hepatitis C viruses. The compiled dataset is comprised of 129,191 protein sequences. In total, 74,468 non-redundant human proteins were retrieved from the NCBI database [46]. The viral protein sequences were subjected to CD-Hit clustering to remove paralogous sequences, with a threshold of 0.6. The resultant 68,322 non-paralogous viral protein sequences were utilised for downstream analysis.

### 2.2. Sequence Similarity Search

The BLASTp program was used to compare 68,322 non-paralogous viral proteins with the human proteome. A total of 24 viral-human homolog proteins were identified using comparative sequence analysis based on a bit-score ≥100, query coverage ≥60, percent identity ≥50, and an E-value of 1 × 10^−6^ (Table 1).

### 2.3. Pathway Enrichment Analysis

The 24 viral protein sequences homologous to human proteins were subjected to pathway enrichment analysis using the PathDIP v4.0.7.0 database [47] for functional annotation. The metabolic pathways were manually compared to identify the proteins involved in the host and virus-shared pathways as well as virus-specific pathways. These pathways were filtered based on a *p*-value of <0.001 to identify significant hits (Appendix A). Thirteen viral proteins were found to share pathways with nine human proteins, whereas eleven viral proteins were involved in virus-specific pathways. Most of the proteins are found to be involved in the autoimmune disease pathways, including cells and molecules involved in the local acute inflammatory response, TNF-related weak inducer of apoptosis (TWEAK) signaling, interleukin-11 signaling, p53 signaling, and inflammation mediated by chemokine and cytokine signaling.

### 2.4. Epitope Prediction

The immunogenic nature of proteins and peptides gives them the potential to bind to the MHC molecules with high binding affinities. MHC-I molecules are found in virtually all nucleated cells and precisely represent endogenous proteins or antigens processed by the cytosolic pathway, which represents cytotoxic T lymphocytes (CTLs). Exogenous antigens, usually the surface proteins of pathogens, are processed by endocytic processes and presented to T lymphocytes or CD4+ T cells [48]. Several algorithms and computational biology resources are available for predicting antigenic epitopes. In silico analysis revealed that pathogenic peptides exhibiting homology with their human counterparts have a significant binding capacity to MHC class-II molecules. Human host proteins with sequence homology to viral peptides may increase the susceptibility to autoimmunity. Epitopes with a binding score of IC_50_ ≤ 500 nM were defined as HLA binders. The 23 most common HLA-DP, HLA-DQ, and HLA-DR alleles were utilized to predict the promiscuity of the mimicking peptides. Peptides with IC_50_ values ≤ 50 nM were speculated to be strong HLA-binders [45]. The epitopes with 15-mer peptide length and 9-mer core residues were prioritized downstream based on IC_50_ values ≤ 50 nM (Figure 1). The analyses eventually identified several promiscuous MHC class-II binding epitopes that demonstrated high binding affinity for all HLA alleles, with an IC_50_ < 50 nM. Several of these homolog epitopes were found to bind promiscuously to several HLA alleles, including HLA-DPA1*02:01/DPB1*05:01, HLA-DQA1*01:02/DQB1*06:02, HLA-DRB1*01:01, HLA-DPA1*01:03/DPB1*03:01, and HLA-DQA1*05:01/DQB1*03:01. The human leukocyte antigen DPB1 (HLA-DPB1) allele has been shown to influence the susceptibility and severity of rheumatoid arthritis. Numerous autoimmune diseases have been linked to HLADRB1-DQA1-DQB1 haplotype components encoded by HLA class-II alleles, including type-1 diabetes, Graves’ disease, and RA [49]. Likewise, the DRB1*01:01 allele has been associated with autoimmune diseases such as rheumatoid arthritis [50]. The ability of the conserved regions to bind to class-II HLA alleles was assessed by docking the lead epitopes in the binding groove of HLA-DRB1, an allele known to predispose individuals to rheumatoid arthritis [51]. SLE and SS are associated with the DRB1*03:01 allele, whereas autoimmune hepatitis (AIH) and RA are reported in association with the DRB1*04:05 allele [52].

### 2.5. Molecular Mimicry Prediction of Viral-Human Homolog Epitopes

The trRosetta server [53] was used to estimate the three-dimensional structures of the viral-human homologous peptides. The resultant 3D structure’s root mean square deviation (RMSD) was determined using the TM-Sore web server with a threshold of 0–1. Viral-human homologous peptides with low RMSD values depicted a significant degree of structural mimicry. These homologous peptides were predicted to be the top candidates that may induce autoimmune diseases. Some peptides exhibited higher RMSD values than the set threshold; nevertheless, their structural resemblance was observed and prioritized.

### 2.6. Molecular Docking

Molecular docking analysis was performed to evaluate the MHC-II binding promiscuity of human and virus-mimicking peptides. Protein-protein rigid body docking was performed using the ClusPro server based on sampling billions of confirmations, RMSD-based clustering, and energy minimization-based structural refinement [54]. Human and virus-mimicking peptides were subjected to molecular docking with human HLA (PDB ID: 5JLZ) and TLR4 (PDB ID: 3FXI) with default parameters. The docking results revealed that the homologous peptide epitopes interact feasibly with HLA and TLR4 receptor epitope-binding sites and develop multiple molecular interactions. The overall docking scores of these peptides within the immune receptors’ antigen-binding sites were found to be significant (Figure 2; Table 2). Peptide fragments with the lowest energy scores showed the highest binding affinities (Appendix A). The molecular interactions of the top prioritized viral-human identical epitope with the receptor protein-interacting residues are shown in Appendix A.

## 3. Discussion

Autoimmune disorders (AID) are caused by an abnormal immune response that fails to distinguish between self and non-self antigens [1,2]. Molecular mimicry is based on the possibility that T- and B-cell antigenic determinants of pathogens may have a counterpart in the host and potentially cause autoimmunity [55]. We implemented an immunoinformatic platform to analyze a wide range of human-infecting viral proteins against the human proteome to prioritize the top hit epitopes that share sequence and 3D structural similarities. The top viral-human homolog peptide epitopes might be valuable targets to examine in association with autoimmune disorders, based on the concept of molecular mimicry. The analysis prioritized 13 viral peptides that showed significant amino acid sequences and 3D structural homologies with various human protein fragments. This includes one viral-human identical epitope that exhibited the highest docking score within the HLA and TLR4 immune receptors. Many of these homologous peptides exhibit promiscuous binding affinities to several MHC class-II molecules, which may provoke autoimmune disorders in humans.

Among the top viral-human mimic candidates, the human herpesvirus and poxvirus proteins exhibited significant 3D structural and sequence homology with human host proteins (Appendix A). Several studies have reported that herpes simplex virus (HSV) is found in active plaques in the postmortem MS brain tissues of patients [56]. Additionally, the HSV-1 infection causes viral gene products to cause neural progenitor cells to undergo apoptosis [57]. The current molecular mimicry findings may help explain the molecular mechanisms underlying the onset of such an autoimmune disorder [54]. A study conducted by Bradshaw et al. (2015) reported herpes simplex virus 1 (HSV1)-induced encephalitis in association with voltage-gated calcium channel autoimmunity through a molecular mimicry mechanism [58].

Herpesvirus infections, particularly Epstein-Barr virus (EBV) infections, have been reported in many studies in association with several major autoimmune diseases [40,59]. We identified epitopic peptide homology between human interleukin-10 (IL-10) and gammaherpesvirus 4 (HHV4). Human IL-10 is a major immune-regulatory cytokine that acts as a potent anti-inflammatory agent, affects a variety of immune cells, and prevents excessive tissue damage by inflammation [60]. Cytokines play a crucial role in the pathogenesis of autoimmune diseases. IL-10 is involved in the pathogenesis of autoimmune diseases, including RA, diabetes, and SLE [61,62]. We also noticed that the gamma herpesvirus 8 (HHV-8) ORF70 protein showed structural and sequence homology with the human TS peptide, and the HHV-8 ORF2 peptide is homologous to the human dihydrofolate reductase (DHFR) peptide. Early studies reported the presence of nine HHV-8 ORF gene products associated with autoimmunity and shared significant homology with human cellular proteins, including the HHV-8 TH and dihydrofolate reductase (DHFR) [63]. The human gammaherpes virus has been reported to be involved in the different systemic autoimmune diseases (SAD), where antibodies against ORFK8.1 were detected in SLS, SS, and vasculitis patients [64].

Significant sequence and structural homology was observed between cercopithecine betaherpesvirus 5 (CHV5) and human peptides in the current study. The prostaglandin G/H synthase 2 of CHV5 showed homology with the human prostaglandin G/H synthase 2 peptide. Human prostaglandin G/H synthase 2 (PTGS2) is involved in the biosynthesis of fibrous tissues, which eventually regulate immune responses during inflammation [65]. Likewise, the peptides of chemokine vCXCL7 from CHV5 showed sequence and 3D structural similarities to the epitopic peptides of the human growth-regulated alpha protein and C-X-C motif chemokine 2. Chemokines are primarily involved in leukocyte recruitment to sites of inflammation and have been reported to contribute to angiogenesis, tumor growth, and organ sclerosis [66].

The current study inferred that various strains of the Poxviridae family share substantial sequence and structural similarities with immunogenic peptides of human proteins. Viruses secrete an array of virus-encoding soluble cytokine receptors or cytokine analogs that act as molecular decoys to inhibit the activity of host cytokines. The variety of poxvirus gene products systematically sabotage essential components of the inflammatory response and manipulate various intracellular signal transduction pathways that initiate proinflammatory responses. Numerous poxvirus genes that interfere with these pathways exhibit striking similarities with host immune system genes [67]. In our analysis, NY_014 poxvirus, monkeypox virus, cowpox virus, akhmeta virus, molluscum contagiosum virus, and vaccinia virus epitopes demonstrated striking resemblance to human proteome origin epitopes.

Metabolic pathways control lineage determination and immune system function, thereby affecting the onset of autoimmune disorders [68]. The majority of host-virus homologous proteins are involved in some of the most important autoimmune disease pathways, such as TNF-related weak inducer of apoptosis (TWEAK) signaling, cellular components involved in local acute inflammatory responses, IL-11 and IL6 signaling pathways, generic transcription pathways, p53 signaling, ataxia-telangiectasia mutated (ATM) signaling, and inflammation mediated by chemokine and cytokine signaling pathways.

The findings of the current study are based on immunoinformatic platforms and constrained prediction methods that may be uncertain due to conventional benchmarking, and a lack of precise datasets. The virus-human molecular mimic candidates prioritized in the current study, therefore, need additional validation via clinical and experimental approaches.

## 4. Materials and Methods

Structural bioinformatics and comparative sequence analysis platforms were followed in this study. The methodological workflow is shown in Figure 3.

### 4.1. Non-Paralogous Viral and Human Proteome Sequence Retrieval

Complete proteome sequences of all human pathogenic viruses were retrieved from the Virus Pathogen Database and Analysis Resource (ViPR) [69]. Redundant sequences were removed, and non-paralogous sequences were acquired using CD-HIT with a threshold of 0.9 (90% sequence similarity) for further analysis. CD-HIT uses a short-read filtering approach to cluster protein sequences with low redundancy [70]. Non-redundant human proteomic sets were retrieved from the NCBI database.

### 4.2. Sequence Similarity Search

A standalone version of the Basic Local Alignment Search Tool (BLAST) was used to compare the viral and human protein sequences [71]. Comparative sequence analyses using BLASTp were performed based on a cutoff e-value of 10^−6^ and a threshold of ≥100 bit score, ≥60% query coverage, and ≥50% sequence identity.

### 4.3. Metabolic Pathway Enrichment

The Path-DIP server was used for protein pathway enrichment analysis with a *p*-value <0.0001 and other default parameters. The Path-DIP database is a curated reference of signaling cascades in human and non-human species that includes core pathways from major curated pathway databases as well as pathways predicted using orthology and physical protein interactions. The Path-DIP provides access to both computationally predicted and experimentally confirmed protein–protein interactions (PPIs) [47].

### 4.4. Epitope Candidate Prediction

The viral-human homolog protein epitopes bound to the MHC-II were predicted using the Immune Epitope Database (IEDB) [72]. The NN-align-2.2 (NetMHCII-2.2) approach predicted binding peptides with multiple human HLA class II alleles, DR, DQ, and HLA-DP, with half-maximal inhibitory concentration (IC_50_) cutoff values of less than 50 nM. Peptides with IC_50_ values less than 50 nM were considered highly compatible binders. The enriched proteins in the metabolic pathways and their respective homologs were subjected to IEDB by selecting HLA class-II binding alleles (Table 3).

### 4.5. Molecular Modeling and Docking Analyses

The 3D structures of the top viral-human homolog peptide epitopes were docked against the human leukocyte antigen (HLA) and toll-like receptor 4 (TLR4) immune receptors. The protein 3D structure data for HLA and TLR4 were acquired from the Protein Data Bank (PDB) via the 5JLZ and 3FXI PDB IDs, respectively. The peptide epitope 3D structures were modeled using the UCSF Chimera programme [73]. The ClusPro server was used to calculate the binding potential of the epitopic peptides to human HLA and TLR4 immune receptors. The ClusPro server generates different models based on binding energies [54].

### 4.6. Structural Mimicry Prediction

The trRosetta server was used to predict the 3D structures of the peptides. The trRosetta server (transform-restrained Rosetta) is a web-based tool that accurately predicts protein and peptide structures. trRosetta is one of the most accurate methods for estimating the 3D structure of molecules via ab initio-based simulations [74]. The 3D structure prediction process in trRosetta is based on energy minimization, with constraints derived from the predicted inter-residue distance and orientations [53]. A template modeling (TM) score was used to measure the structural similarity of the peptides based on the root mean square deviation (RMSD).

## 5. Conclusions

Comparative sequence analyses and immunoinformatics approaches were employed in the current study to uncover sequence- and structure-based molecular mimics of viral-human proteins as possible autoimmune candidates. BLASTp analysis identified 24 viral proteins with significant sequence homology to human host proteins. Biological pathway enrichment analysis revealed the involvement of viral-human homolog proteins in a variety of human metabolic pathways. Immune epitope prediction analysis inferred that the viral-human homolog proteins shared 13 promising T-cell epitopes, suggesting promiscuous binding to human HLA class II alleles. The experimental validation of these proteins and their top-hit host-pathogen homolog epitopes may explain the virus-based autoimmune diseases in humans.

## Figures and Tables

**Figure 1 pathogens-11-01362-f001:**
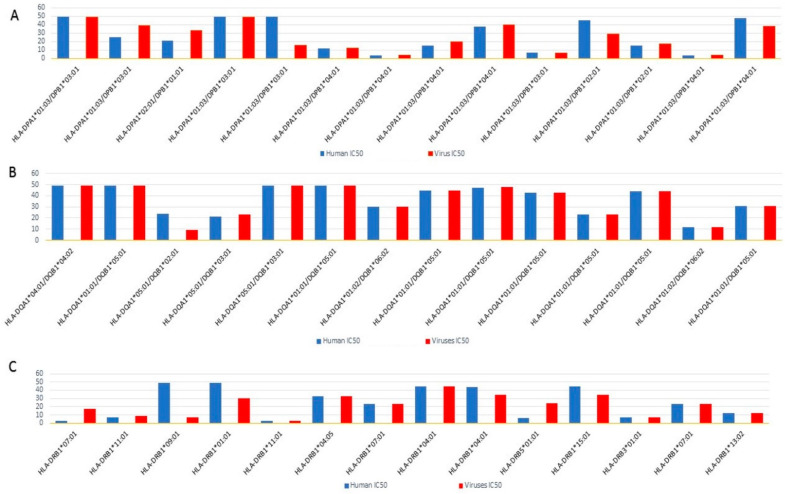
The viral-human mimicking epitopes’ binding potential against HLA alleles, (**A**) HLA-DP; (**B**) HLA-DQ; and (**C**) HLA-DR. The plot represents the total number of binders for each allele. Strong binders (IC_50_ ≤ 50), weak binders (IC_50_ ≤ 500), and non-binders (IC_50_ > 500) were assigned to each fragment based on their expected IC_50_ values. The y-axis represents the IC_50_ values.

**Figure 2 pathogens-11-01362-f002:**
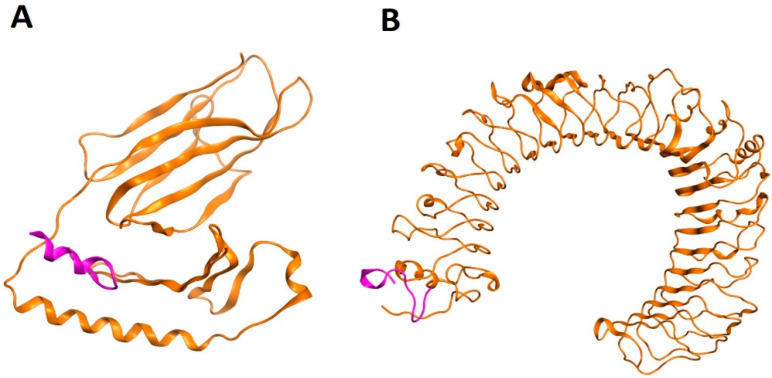
Molecular docking of the viral-human identical epitope (mentioned in S. No. 1, Table 2) with HLA (**A**) and TLR4 (**B**) receptors based on lowest binding energy. The purple colour indicates the top-prioritized peptide epitope, and the orange colour indicates the receptor proteins.

**Figure 3 pathogens-11-01362-f003:**
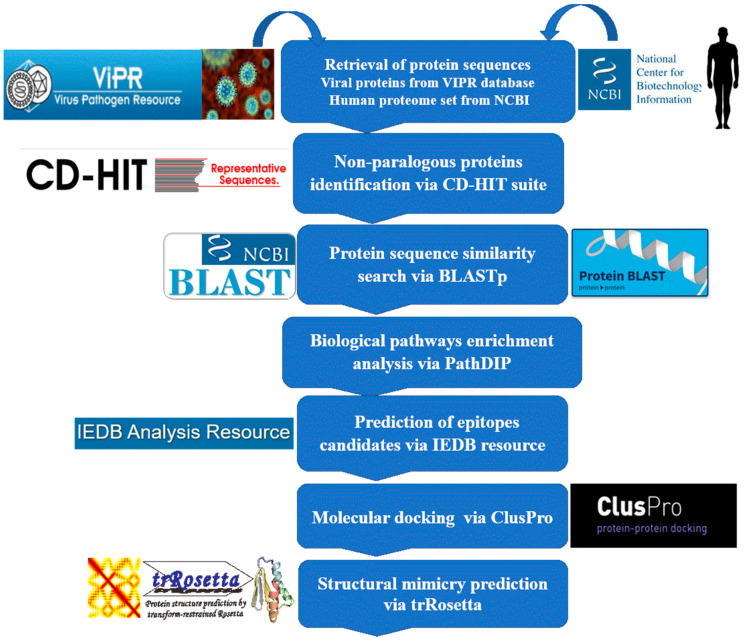
Systematic workflow diagram of autoimmune mimicry of viruses with human hosts, followed by T-cell epitope prediction, structural analysis, and molecular docking analysis.

**Table 1 pathogens-11-01362-t001:** The top viral and human homolog proteins hits acquired from comparative proteome sequence analysis.

S. No	Query Sequence IDs	Subject Sequence IDs	Score	Query Coverage	Percent-Identity	E-Value
**1**	gb:AXN75085	sp|P23921|RIR1_HUMAN	3148	98	75.297	0
**2**	gb:AST09466	sp|P23921|RIR1_HUMAN	3130	100	73.106	0
**3**	gb:AEV80548	sp|P35354|PGH2_HUMAN	2452	98	73.729	0
**4**	gb:AAY97564	sp|P49916|DNLI3_HUMAN	1700	98	55.124	0
**5**	gb:AST09563	sp|P49916|DNLI3_HUMAN	1594	99	51.852	0
**6**	gb:AZY90656	sp|P31350|RIR2_HUMAN	1412	95	80.625	0
**7**	gb:AST09433	sp|P31350|RIR2_HUMAN	1399	99	80.312	0
**8**	gb:QCA43223	sp|P04818|TYSY_HUMAN	1162	95	71.579	2.94 × 10^−161^
**9**	gb:BBA90853	sp|P04818|TYSY_HUMAN	1153	89	69.333	2.63 × 10^−159^
**10**	gb:AQY16903	sp|Q9H2F3|3BHS7_HUMAN	834	99	50.559	9.70 × 10^−110^
**11**	gb:ADZ29327	sp|Q9HC24|LFG4_HUMAN	833	100	71.849	1.56 × 10^−113^
**12**	gb:AZT86284	sp|P07203|GPX1_HUMAN	831	83	84.153	6.52 × 10^−114^
**13**	gb:AXN75107	sp|P04183|KITH_HUMAN	656	97	69.006	1.45 × 10^−87^
**14**	gb:AST09487	sp|P04183|KITH_HUMAN	625	98	64.205	6.60 × 10^−83^
**15**	gb:QCF48225	sp|P22301|IL10_HUMAN	599	95	81.287	6.42 × 10^−80^
**16**	gb:ABD28857	sp|P00374|DYR_HUMAN	498	86	50	9.21 × 10^−64^
**17**	gb:AAY97032	tr|H0YNW5|H0YNW5_HUMAN	477	90	64.964	7.74 × 10^−62^
**18**	gb:AUL80434	tr|A0A0C4DGL3|A0A0C4DGL3_HUMAN	394	92	55.882	1.42 × 10^−49^
**19**	gb:AUL80132	sp|Q8IV08|PLD3_HUMAN	377	85	50.595	6.87 × 10^−43^
**20**	gb:AAY97407	sp|Q9HC24|LFG4_HUMAN	367	97	63.248	9.72 × 10^−45^
**21**	gb:AUL80484	sp|P23921|RIR1_HUMAN	306	86	57.143	2.35 × 10^−33^
**22**	gb:AUL80431	tr|A0A3B3ITT3|A0A3B3ITT3_HUMAN	235	86	54.167	2.36 × 10^−25^
**23**	gb:AEV80662	sp|P09341|GROA_HUMAN	208	71	65.079	1.00 × 10^−22^
**24**	gb:AEV80661	sp|P19875|CXCL2_HUMAN	180	60	55.932	2.11 × 10^−18^

**Table 2 pathogens-11-01362-t002:** Molecular docking scores of human and virus-mimicking peptides docked with the human HLA and TLR4 receptors. Viral-human homolog epitopes capable of binding to human host MHC class-II (“:” denotes the distance between residue pairs).

S. No	Human Proteins	Human Peptides	Docking Score in Human HLA	Docking Score in Human TLR4	Virus Proteins	Virus Peptides	Docking Score in Human HLA	Docking Score in Human TLR4	Structural Mimicry	RMSD Value
**1**	Ribonucleoside-diphosphate reductase subunit M2	IFFSGSFASIFWLKK	−1105.4	−829.4	CPXV051	IFFSGSFASIFWLKK	−1105.4	−829.4	IFFSGSFASIFWLKK ::::::::::::::: IFFSGSFASIFWLKK	0.00
**2**	Thymidine kinase, cytosolic	STELMRRVRRFQIAQ	−864	−689	thymidine kinase	STELIRRVRRYQIAQ	−853	−596.4	STELMRRVRRFQIAQ ::::::::::::::: STELIRRVRRYQIAQ	0.1
**3**	Ribonucleoside-diphosphate reductase large subunit	RDFSYNYFGFKTLER	−854	−659.7	ribonucleotide reductase large subunit	RDFSYNYFGFKTLER	−854	−659.7	RDFSYNYFGFKTLER ::::::::::::::: RDFSYNYFGFKTLEK	0.1
**4**	Prostaglandin G/H synthase 2	MFAFFAQHFTHQFFK	−962	−742.8	prostaglandin G/H synthase 2	MFAFFAQHFTHQFFK	−962	−742.8	MFAFFAQHFTHQFFK ::::::::::::::: MFAFFGQHFTHQFFR	0.1
**5**	Thymidylate synthase	TKRVFWKGVLEELLW	−962.9	−747	ORF13	TKRVFWRAVVEELLW	−117.6	−738	TKRVFWKGVLEELLW ::::::::::::::: TKRVFWRAVVEELLW	0.2
**6**	Thymidylate synthase	VPFNIASYALLTYMI	−874.7	−613	ORF70	VPFNIASYSLLTYML	−867	−745.6	VPFNIASYALLTYMI :::::::::::: VPFNIASYSLLTYML	0.3
**7**	Dihydrofolate reductase	RPLKGRINLVLSREL	−823	−1070	ORF2	RPLAGRINVVLSRTL	−1087	−782	RPLKGRINLVLSREL ::::::::::::::: RPLAGRINVVLSRTL	0.8
**8**	DNA ligase 3	FVFDCIYFNDVSLMD	−951.7	−861	ATP-dependent DNA ligase	FVFDCIYFNDVSLMD	−951.7	−861	FVFDCIYFNDVSLMD : :::::::::: FVFDCIYFNDVSLMD	0.9
**9**	thymidine kinase	LMRRVRRFQIAQYKC	−863.6	−691	thymidine kinase	LIRRVKRYQIAKYDC	−907	−652	LMRRVRRFQIAQYKC ::::::::::::::: LIRRVKRYQIAKYDC	1
**10**	Ribonucleoside-diphosphate reductase large subunit	AGRRAAGASVATELR	−875.3	−781.9	ribonucleotide reductas	LMSLIAYCQSATELR	−917	−806.5	AGRRAAGASVATELR :::::::::: LMSLIAYCQSATELR	1.0
**11**	Growth-regulated alpha protein	IIYDRDFSYNYFGFK	−661	−602.4	chemokine vCXCL7	IINDRDFSYNYFGFK	−789.5	−580.5	IIYDRDFSYNYFGFK ::::::::: IINDRDFSYNYFGFK	1.0
**12**	C-X-C motif chemokine 2	LLLVAASRRAAGAPL	−763	−781	chemokine vCXCL6	SRLLVATLLGTLLAC	−1006.9	−598	LLLVAASRRAAGAPL ::::::::: SRLLVATLLGTLLAC	1.5
**13**	DNA ligase 3	CLFVFDCIYFNDVSL	−820.4	−742.3	DNA ligase	CLFVFDCLYFDGFDM	−942	−878	CLFVFDCIYFNDVSL ::: ::::::::: CLFVFDCLYFDGFDM	2.1

**Table 3 pathogens-11-01362-t003:** The HLA class-II binding alleles selected in the current study based on an IC_50_ value < 50 nM.

S/No	HLA-DP	HLA-DQ	HLA-DR
1	DPA1*01:03-DPB1*02:01	DQA1*01:02–DQB1*06:02	DRB1*03:01
2	DPA1*02:01–DPB1*05:01	-DQA1*04:01–DQB1*04:02	DRB1*04:04
3	DPA1*03:01–DPB1*04:02	DQA1*05:01–DQB1*03:01	DRB1*07:01
4	DPA1*01:03–DPB1*04:01	DQA1*01:01–DQB1*05:01	DRB1*11:01
5	DPA1*01:03-DPB1*03:01-DPB1*04:01	DQA1*05:01–DQB1*03:01	DRB1*13:02
6	DPA1*02:01-DPB1*01:01	DQA1*03:01–QB1*03:02	DRB3*01:01
			DRB5*01:01
			DRB1*01:01
			DRB4*01:01
			DRB1*04:01
			DRB1*15:01
			DRB1*04:05
			DRB1*11:01
			DRB1*08:02

## Data Availability

All of the relevant data are provided in the form of regular figures, tables, and Appendix A files.

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
