# Peer review of "Molecular Mimicry Analyses Unveiled the Human Herpes Simplex and Poxvirus Epitopes as Possible Candidates to Incite Autoimmunity"

_pathogens, 2022, doi:10.3390/pathogens11111362_

Round 1
Reviewer 1 Report (New Reviewer)
The aim of the study was to implement the immunoinformatics approaches to infer sequence and structure-based molecular mimicry between viral and human proteome data sets. The authors found that the top-hit homolog epitopic peptides belong to the herpes simplex and poxvirus families. According to the authors, these findings could lead to epitopes' biological sequences, and 3D structural-based molecular mimicry may promising to interpret the herpes simplex virus and poxvirus infection-mediated autoimmune disorders in humans.
Minor comments:
The authors must describe in the discussion section the limitations of the study at the methodological level, study design, and establishment of causal relationships or associations, as well as the perspectives of the study.
Author Response
Please see the attachment.

Reviewer 2 Report (New Reviewer)
Summary: The authors of this publication have investigated the molecular mimicry of human and viral proteins. These viral proteins that resemble human proteins can induce autoimmunity. The authors have selected a topic of infectious diseases that is frequently overlooked but which can provide a unique perspective on the disease. This study identified 24 viral proteins with significant host homology. A few homologs of these proteins displayed significant binding to HLA and TLR4 immune receptors.
The article was competently written, and the author selected a pertinent topic. The review and compilation of prior research in the field by the author are nicely done. However, there are a few concerns that should be addressed:
1. Because of the primary emphasis on screening, the article appears to lack focus. The first part should be screening, and the subsequent article should narrow its focus and elaborate more on the primary homolog targets.
2. The article is solely based on in silico data, which is good for screening targets. But the authors need to validate the results further using in vivo and in vitro studies.
3. The protein homologs found in the study should be explained properly with their function in the host/virus.
4. Figure 1: labeling should be more prominent and clearer; it is unreadable at the current font size.
5. Figure 2: is not properly made. The receptor and ligand are not color-coded correctly to be distinguishable. It is not mentioned in the article where the homologs bind on the receptor as well as the interacting residues.
6. Table 2: the protein from which human and virus peptides originated should be clearly mentioned.
In conclusion, the work done in the article is valuable, but there is more work needed in the article for it to be published.
Round 2
Reviewer 2 Report (New Reviewer)
The authors have revised the manuscript, and it can be accepted now.
This manuscript is a resubmission of an earlier submission. The following is a list of the peer review reports and author responses from that submission.
Round 1
Reviewer 1 Report
In the present manuscript, Begum and colleagues performed comparative analyzes of protein sequences of known human infectious viruses and available human proteome data in the UniProt database using different bioinformatics methods. They found 24 viral proteins that have high sequence similarity to human proteins. Prediction of binding to HLA molecules and TLR4 and molecular modeling were also performed. The authors have tried to identify viral epitopes that could induce autoimmune response.
I have some serious concerns and remarks.
1. There is a misunderstanding of the concepts of homologous proteins and molecular mimicry, which leads to misinterpretation of the data. On the one hand, the presence of homology does not need to trigger an autoimmune response. On the other hand, to speak for molecular mimicry, cross-activation of autoreactive T or B cells by viral epitopes must be established. In the present study, there are only bioinformatics analyses and molecular modeling, but not experimental validation or functional tests. It is well known that molecular modeling is far from the reality.
2. Homology of the proteins participating in different metabolic pathways is not related to the autoimmunity.
3. For many of the autoimmune diseases, the T- and B-cell epitopes are identified and well-studied. Suggested 14 viral peptides/epitopes homologous to human proteins presented in Tables 3 and 4 are not among them. It is not known whether they are immunogenic at all. No association with a specific autoimmune disease has been shown or proven.
4. Cancer and Alzheimer's disease (Table 5) are not autoimmune diseases.
5. Section ‘Discussion’ is descriptive and looks more like a review than a discussion of the results obtained against the established and well-known data for the autoimmune diseases.
Minor points:
- Table 2 should be given in the supplementary materials.
- It is not clear what is presented in Figure 1 (Y-axis).
- It is not clear which peptides are used for the molecular docking analyzes in Figure 2. Contact points are not shown or discussed. How many models were generated and which were selected?
- The same is valid for Figure 3. It is not clear which peptides are presented.
- References given in Table 5 are not in the reference list. The peptide/epitope sequences presented in this table are not related/discussed in the cited references.
- Table 3 and Table 4 can be combined into one table.
- What is the difference between the peptides #1 and #2 in Tables 3 and 4?
- How would the authors explain the different Docking Score values obtained for the same peptide (the viral and human peptide sequences are exactly the same) – Table 3, peptides 1(2), 4, 5, and 9?
- The reference list is not prepared according to the style of the journal.
Responses to reviewer’s comments
- There is a misunderstanding of the concepts of homologous proteins and molecular mimicry, which leadsto misinterpretation of the data. On the one hand, the presence of homology does not need to trigger an autoimmune response. On the other hand, to speak formolecular mimicry, cross-activation of autoreactive T or B cells by viral epitopes must be established. In the present study, there are only bioinformatics analyses and molecular modeling, but not experimental validation or functional tests. It is well known that molecular modeling is far from the reality.
Author response- We appreciate the reviewer concern. However, it cannot be denied that host-pathogen molecular mimicry is one of the main cause for autoimmunity diseases in humans. Many studies have been conducted in this regard and quite a valuable literature is available. We would like to present below some references where molecular mimicry is associated with multiple autoimmune disorders in humans caused by various pathogens, and to explain cross activation of autoreactive T & B cells by viral epitopes.
- Smatti MK, Cyprian FS, Nasrallah GK, Al Thani AA, Almishal RO, Yassine HM. Viruses and Autoimmunity: A Review on the Potential Interaction and Molecular Mechanisms. Viruses. 2019 Aug 19;11(8):762. doi: 10.3390/v11080762. PMID: 31430946; PMCID: PMC6723519.
- Cusick MF, Libbey JE, Fujinami RS. Molecular mimicry as a mechanism of autoimmune disease. Clin Rev Allergy Immunol. 2012 Feb;42(1):102-11. doi: 10.1007/s12016-011-8294-7. PMID: 22095454; PMCID: PMC3266166.
- van Langelaar J, Rijvers L, Smolders J, van Luijn MM. B and T Cells Driving Multiple Sclerosis: Identity, Mechanisms and Potential Triggers. Front Immunol. 2020 May 8;11:760. doi: 10.3389/fimmu.2020.00760. PMID: 32457742; PMCID: PMC7225320.
Besides, in the current age of biology where quite a lot of larger biological datasets are available publically due to advancement in the OMICS technologies, the field of molecular modeling, bioinformatics based data analyses has gained a lot of attention. The approaches provide quite a lot of informative hits for understanding the molecular basis of complex diseases. We employed computational biology resources and utilized updated biological databases data to prioritize the top hits viral epitopes that might causes auto-immunity in human. By comparing thousands of viruses and human host proteins, we shortlisted the top hits to facilitate the researchers to key focus during experimentation. Though we have no funding and facilities currently to validate the findings experimentally, but we believe that the information is worthy for scientific community to be followed in future. The information could provide new direction to study the mechanisms of viral-mediated autoimmunity and to identify the biological pathways involved in autoimmunity.
- Homology of the proteins participating in different metabolic pathways is not related to the autoimmunity.
Author response: I appreciate the reviewer concern, however, if you see the results section 2.3, most the viral-human homologs proteins are found to be involved in autoimmune disease pathways, including cells and molecules involved in local acute inflammatory response, TNF related weak inducer of apoptosis (TWEAK) signaling, interleukin-11 signaling, p53 signaling and inflammation mediated by chemokine and cytokine signaling. Besides, it should be noted that these information are based on so far known metabolic pathways. We identified many other viral-human homologs proteins as well, but there biological pathways enrichment information still not available and need to be explored. This information are therefore worthy for scientific community.
- For many of the autoimmune diseases, the T- and B-cell epitopes are identified and well-studied. Suggested 14 viral peptides/epitopes homologous to human proteins presented in Tables 3 and 4 are not among them. It is not known whether they are immunogenic at all. No association with a specific autoimmune disease has been shown or proven.
Author response: We appreciate the reviewer concern, however, it should be noted that we followed a stringent criteria of comparative sequences analyses to call the viral-human homologs epitopic peptides. Thousands of viral pathogens and human host proteins were thoroughly scanned for homology search. The immunogenic nature of shortlisted peptides have been clarified via comprehensive analyses through multiple well-renowned immunoinformatics resources as well as with subsequent molecular docking analyses against human immune receptors. As per our literature review, we still believe that viral based autoimmunity diseases molecular investigation is quite shallow and the current viral-human homolog peptides identified are therefore promising, top hit molecular entity that may provoke the autoimmunity.
- Cancer and Alzheimer's disease (Table 5) are not autoimmune diseases.
Author response: We thanks the reviewer. Keep in view this concern, we re-analyzed the pathways enrichment data. Few of the pathways enrichment results were mistakenly interpreted and therefore this particular portion has been completely revised. During repeat analyses, none of the top viral-human homolog proteins or peptides get enriched in the Cancer and Alzheimer's diseases pathways. We appreciate the reviewer and apologies for the mistake.
- Section ‘Discussion’ is descriptive and looks more like a review than a discussion of the results obtained against the established and well-known data for the autoimmune diseases.
Author response- The discussion portion is rewritten as per reviewer’s suggestion. All the unnecessary additional information has been discarded. Only the information supporting our results are included.
Minor points:
- Table 2 should be given in the supplementary materials.
Author response- Thanks for the suggestion. Table 2 has been added to supplementary materials as Table S1.
- It is not clear what is presented in Figure 1 (Y-axis).
Author response- The Y-axis in figure 1 represents the IC50 values. This has now been explained in the revised manuscript figure 1. Thanks.
- It is not clear which peptides are used for the molecular docking analyzes in Figure 2. Contact points are not shown or discussed. How many models were generated and which were selected?
Author response- Thanks for the concern. Actually, the top most homolog peptide shown in S. no. 1, Table 2 was selected to dock with the HLA and TLR4 receptors. This has now been explained now in figure 2 legend of the revised manuscript. Docking was performed via ClusPro server (https://cluspro.org). It is a widely used tool for protein-protein and protein-peptide docking. ClusPro employs six different energy functions depending on the type of proteins. Docking with each energy parameter set results in ten models defined by centers of highly populated clusters of low energy docked structures. Among the ten models, we selected the most stable one, i.e. least energy model. The top viral-human homolog peptides were found to dock within the epitope binding regions of TLR4 and HLA alleles feasibly and were found to develop multiple molecular interaction, including H-bonding with the receptors. Actually, the interaction of the peptide epitope was though energetically favorable and feasible in both the receptors, however the residues contact and interactions pattern were found different as the molecular environment of the antigen binding sites of the two receptors (TLR4 & HLA allele) is distinct. The interpretation has been revised in the manuscript.
- The same is valid for Figure 3. It is not clear which peptides are presented.
Author response: Thanks. It has been revised and clarified in the figure 3 legend.
- References given in Table 5 are not in the reference list. The peptide/epitope sequences presented in this table are not related/discussed in the cited references.
Author response: We thanks the reviewer. We checked all the data analyses. In some cases the analyses were re-performed. We are extremely sorry about some incorrect information previously given in Table 5. This all has been discarded and the results obtained during re-analyses are discussed in the main manuscript.
- Table 3 and Table 4 can be combined into one table.
Author response: The manuscript revised and the Table 3 & Table 4 are combined as per suggestion and renamed as Table 2.
- What is the difference between the peptides #1 and #2 in Tables 3 and 4?
Author response: Actually these were the identical peptides acquired from two different proteins. This has been revised and showed once in the table.
- How would the authors explain the different Docking Score values obtained for the same peptide (the viral and human peptide sequences are exactly the same) – Table 3, peptides 1(2), 4, 5, and 9?
Author response: The docking scores are exactly the same for the same receptors. The difference in the docking score in the last manuscript version was because of using different HLA molecules. To avoid misunderstanding, the same receptor has now used in the re-analyses and the entire manuscript has been revised in this regard.
- The reference list is not prepared according to the style of the journal.
Author response: Thanks. This has been revised and the reference list is prepared as according to the style of the journal.